# Unlabelled Sample Compression Schemes for Intersection-Closed Classes and Extremal Classes

**J. Hyam Rubinstein**
School of Mathematics & Statistics
University of Melbourne
Parkville, VIC 3052, Australia
`joachim@unimelb.edu.au`

**Benjamin I. P. Rubinstein**
School of Computing & Information Systems
University of Melbourne
Parkville, VIC 3052, Australia
`brubinstein@unimelb.edu.au`

## Abstract

The sample compressibility of concept classes plays an important role in learning theory, as a sufficient condition for PAC learnability, and more recently as an avenue for robust generalisation in adaptive data analysis. Whether compression schemes of size $O(d)$ must necessarily exist for all classes of VC dimension $d$ is unknown, but conjectured to be true by Warmuth. Recently Chalopin, Chepoi, Moran, and Warmuth (2018) gave a beautiful unlabelled sample compression scheme of size VC dimension for all maximum classes: classes that meet the Sauer-Shelah-Perles Lemma with equality. They also offered a counterexample to compression schemes based on a promising approach known as corner peeling. In this paper we simplify and extend their proof technique to deal with so-called extremal classes of VC dimension $d$ which contain maximum classes of VC dimension $d - 1$. A criterion is given which would imply that all extremal classes admit unlabelled compression schemes of size $d$. We also prove that all intersection-closed classes with VC dimension $d$ admit unlabelled compression schemes of size at most $11d$.

## 1  Introduction

Littlestone and Warmuth (1986) initiated the study of sample compression schemes as an alternative characterisation of probably approximately correct (PAC) learnability, to finite Vapnik-Chervonenkis (VC) dimension. Informally a sample compression scheme for a concept class comprises a compressor and reconstructor: while the compressor represents samples labelled by the class as a labelled or unlabelled subsample, the reconstructor must be able to recover a concept consistent with the original labelled sample given only this representation. Littlestone and Warmuth (1986) demonstrated a remarkably simple proof of PAC learnability for any concept classes with labelled compression schemes provided compressed representations have bounded size, constant in the original sample size. They conjectured that bounded labelled compression schemes should also be necessary for PAC learnability—completing the characterisation—specifically that all VC-$d$ concept classes (those having VC dimension $d$) have labelled compression schemes of size $d$. This conjecture, for size $O(d)$, remains open now more than 35 years later, and stands as one of the oldest open problems in learning theory. In addition to proving PAC generalisation bounds (von Luxburg, Bousquet, and Schölkopf, 2004; Langford, 2005), compression has found important connections to generalisation in the adaptive data analysis setting (Cummings, Ligett, Nissim, Roth, and Wu, 2016).

While Littlestone and Warmuth (1986) conjectured the existence of labelled schemes of size $d$ for all VC-$d$ classes, Warmuth (2003) later relaxed this to $O(d)$, while Kuzmin and Warmuth (2007) conjectured size $O(d)$ *unlabelled* compression schemes. (In unlabelled schemes, the compressor's representations omit labels; such schemes of size $k$ imply labelled schemes of size $k$.) Indeed VC-$d$ classes are known that cannot be unlabelled compressed to size $d$ (Pálvölgyi and Tardos, 2020).

36th Conference on Neural Information Processing Systems (NeurIPS 2022).

Notable progress towards resolving the sample compression conjecture has involved compressing families of concept classes. Shortly after the original conjecture Floyd (1989) proved that all maximum concept classes $\mathcal{C}$ (those meeting the Sauer-Shelah-Perles Lemma with equality) can be label compressed to size VC$(\mathcal{C})$ (Floyd and Warmuth, 1995). Chalopin et al. (2018) recently proved the same bound but with unlabelled compression, while also providing an elegant counter example to the same result for so-called corner peeling schemes. This shows that the constructions of corner peeling schemes by Kuzmin and Warmuth (2007) and Rubinstein and Rubinstein (2012) are incorrect. Moran and Yehudayoff (2016) proved that all VC-$d$ classes can be label compressed to size exponential in $d$. Recently Pálvölgyi and Tardos (2020) offered examples of VC classes which cannot admit unlabelled compression schemes of size $d$.

Moran and Warmuth (2016) discover labelled compression schemes for extremal classes, which contain cubes for any coordinates shattered by the class, and generalise maximum classes. It is unknown if all extremal classes admit unlabelled compression schemes of size $O(d)$. In this paper, we give a criterion on extremal classes to admit unlabelled compression schemes of size VC dimension

Note that in Chalopin et al. (2018), a beautiful counterexample is given to the possibility of all maximum classes having corner peeling unlabelled compression schemes of size VC dimension. This shows that the constructions of corner peeling schemes in Kuzmin and Warmuth (2007) and Rubinstein and Rubinstein (2012) are incorrect.

A promising approach to the general conjecture—and a strong motivation for compressing special families of concept classes–is via embedding, as a compression scheme restricts to any subclass. So, there is interest in embedding general VC-classes into larger classes with good compression properties. Rubinstein, Rubinstein, and Bartlett (2015) proved that VC-$d$ classes cannot always be embedded in maximum classes of VC dimension less than $2d-1$. Embedding VC-$d$ classes into maximum classes of VC-dimension $O(d)$ could still achieve compression to size $O(d)$ in general.

Intersection-closed classes—those closed under coordinate-wise multiplication—are another promising focus for compression, as any class can be embedded into an intersection-closed class. In this paper, we show that any intersection-closed class can be embedded into an extremal intersection-closed class with the VC dimension increasing from $d$ to $11d$. These extremal intersection-closed classes have VC-sized unlabelled compression schemes by Chalopin et al. (2018). As such, we establish for the first time that intersection-closed classes can be unlabelled compressed to size linear in their VC dimension. We offer a short proof of the result by Chalopin et al. (2018) that maximum classes have VC dimension $d$-size unlabelled compression schemes and extend this to extremal classes which contain maximum classes of VC dimension $d-1$. Finally we give an efficient method to compute the VC dimension of the intersection closure of a class. In particular, this could be useful in searching for collections of classes for which the VC dimension does not increase by much when passing to the intersection closure and then constructing the above unlabeled compression scheme.

## 2 Preliminaries

Given $n \in \mathbb{N}$ we call a subset of the binary $n$-cube $\mathcal{C} \subseteq \{0,1\}^n$, a *concept class* with elements $v \in \mathcal{C}$ *concepts*. These come from restrictions of classifier families, on samples of $n$ instances.[1] We equate each instance in $[n] = \{1, \ldots, n\}$ with *coordinates*, *axes* or *colours* of the $n$-cube. An important combinatorial parameter is the VC dimension, which is used to bound concept class cardinality.

**Definition 2.1** (Vapnik and Chervonenkis, 1971). The *VC dimension* of concept class $\mathcal{C} \subseteq \{0,1\}^n$ is defined as VC$(\mathcal{C}) = \max\{|I| : I \subseteq [n], \text{proj}_I(\mathcal{C}) = \{0,1\}^{|I|}\}$, where $\text{proj}_I(\mathcal{C}) = \{(v_i)_{i \in I} : v \in \mathcal{C}\}$ is the set of *coordinate projections* of the concepts of $\mathcal{C}$ on coordinates $I \subseteq [n]$. $\mathcal{C}$ is said to *shatter* $I$ when the projection $\text{proj}_I(\mathcal{C}) = \{0,1\}^{|I|}$ is onto the $|I|$-cube.

**Lemma 2.2** (Sauer-Shelah-Perles Lemma, Vapnik and Chervonenkis, 1971; Shelah, 1972; Sauer, 1972). *The cardinality of any concept class $\mathcal{C} \subseteq \{0,1\}^n$ is bounded by $|\mathcal{C}| \leq \sum_{i=0}^{\text{VC}(\mathcal{C})} \binom{n}{i}$.*

Concept classes meeting equality in Sauer's Lemma are called *maximum* (Welzl, 1987). A cubical–complex structure can be put on concept classes $\mathcal{C} \subseteq \{0,1\}^n$. To begin, each $v \in \mathcal{C}$ is a vertex, and

---

[1]Consider family of classifiers $\mathcal{F}$ mapping input domain $\mathcal{X}$ to labels $\{0,1\}$. On sample $\mathbf{x} \in \mathcal{X}^n$, the family is represented by induced concept class $\mathcal{C} = \{(f(x_1), \ldots, f(x_n) : f \in \mathcal{F}\}$.

pairs of Hamming-1 separated vertices are edges, of the *one-inclusion graph* of $\mathcal{C}$ (Haussler, Littlestone, and Warmuth, 1994). Each higher-order cube is all points varying over a set of coordinates, the cube's *colours*; a cube's common values on non-colour coordinates make up its *anchor*. (Equivalently $d$-cubes are pre-images of points under a projection from the binary $(m+d)$-cube to the binary $m$-cube.) Any $d$-maximum class has a special structure as a complete collection of $d$-cubes:

**Definition 2.3** (Rubinstein, Bartlett, and Rubinstein, 2009)**.** A set of $d$-cubes $\mathcal{S}$ of cardinality $\binom{n}{d}$ is called $d$-*complete* if for all $I \subseteq [n]$ of cardinality $d$, there exists $c_I \in \mathcal{S}$ with $\mathrm{proj}_I(c_I) = \{0,1\}^d$.

Another property of a concept class $\mathcal{C}$ is of being *shortest-path closed*—also known as *isometric* (Chalopin et al., 2018)—where every pair of points in $\mathcal{C}$ has a shortest path contained in $\mathcal{C}$ (Kuzmin and Warmuth, 2007; Rubinstein and Rubinstein, 2012). We will be especially interested in *extremal* classes—also known as *ample* classes (Chalopin et al., 2018).

**Definition 2.4.** A class $\mathcal{C} \subseteq \{0,1\}^n$ is *extremal* if whenever $\mathcal{C}$ shatters some $I \subseteq [n]$, then there is a subcube $c \subseteq \mathcal{C}$ which maps bijectively to the $|I|$-cube via $\mathrm{proj}_I(\cdot)$. That is, $\mathrm{proj}_I(c) = \{0,1\}^{|I|}$.

All maximum classes are extremal and extremal classes are shortest-path closed (Moran and Warmuth, 2016), while the complements of extremal classes are themselves extremal (Bandelt, Chepoi, Dress, and Koolen, 2006; Bollobás and Radcliffe, 1995; Lawrence, 1983). As an example, a VC one dimensional maximum (respectively extremal) class is a tree with one (respectively at most one) edge for each colour. An extremal counter-part to Sauer's Lemma was independently discovered by Pajor (1985), Bollobás and Radcliffe (1995), Dress (1997), and Anstee, Rónyai, and Sali (2002).

**Lemma 2.5** (Sandwich Lemma)**.** *For any extremal class $\mathcal{C}$, its cardinality is equal to both the number of shatterings of $\mathcal{C}$ and the number of cube types (an equivalence class of cubes which all have the same set of colours) in $\mathcal{C}$.*

Extremal and maximum classes have an elegant recursive structure that is best studied by inductive arguments. One such tool is the projection operator $\mathrm{proj}_I(\mathcal{C})$. The *reduction* $\mathcal{C}^I \subseteq \mathrm{proj}_I(\mathcal{C})$ is the subset of points with non-unique pre-images under the projection, while the *tail* is the subset of $\mathrm{proj}_I(\mathcal{C})$ with unique pre-image under the projection (Welzl, 1987; Kuzmin and Warmuth, 2007).

Unlabelled compression schemes are defined by *representation mappings*. A party known as the *compressor* uses a representation mapping to summarise a $\mathcal{C}$-labelled sample of any size with a small unlabelled subset of instances. A second party, the *reconstructor*, can invert this representation thanks to condition 1, to a concept that is consistent with the original labelled sample, thanks to condition 2, (Chalopin et al., 2018).

**Definition 2.6** (Kuzmin and Warmuth, 2007)**.** Mapping $r$ is a *representation mapping of size $k < n$*, also known as an *unlabelled compression scheme of size $k$*, of class $\mathcal{C} \subseteq \{0,1\}^n$ if it satisfies:

1. $r$ is an injection between $\mathcal{C}$ and subsets of $[n]$ of size at most $k$; and
2. (non-clashing) $\mathrm{proj}_{r(v) \cup r(v')}(v) \neq \mathrm{proj}_{r(v) \cup r(v')}(v')$ for all $v \neq v' \in \mathcal{C}$.

## 3 The intersection closure operator

Examples of intersection-closed classes include axis-aligned hyperrectangles in $\mathbb{R}^m$, monomials, which are unions of orthogonal subcubes, and any union of orthogonal intersection-closed subclasses (Moran and Warmuth, 2016). Closed-below classes (Rubinstein et al., 2009) are intersection closed and extremal. All extremal classes of VC dimension 1 are intersection closed, so long as the origin is located in the class (Moran and Warmuth, 2016).

Note the property of being intersection closed depends on the choice of origin.

We will use the notation $v \odot w$ to denote the intersection of the vertices $v, w$. This is just the operation of bitwise Boolean multiplication $v \odot w = (v_1 w_1, \ldots, v_n w_n)$. We can likewise let $u^1 \odot \ldots \odot u^k = (\prod_{i=1}^k u_1^i, \ldots, \prod_{i=1}^k u_n^i)$ denote the (iterated) intersection of vertices $u^1, \ldots, u^k$.

**Definition 3.1.** A class $\mathcal{C}$ in the binary $n$-cube is called *intersection closed* if $v \odot w \in \mathcal{C}$, whenever $v, w \in \mathcal{C}$. Let $\overline{\mathcal{C}} = \odot(\mathcal{C}) = \{u^1 \odot \ldots \odot u^k) : k \leq n, \{u^1, \ldots, u^k\} \subseteq \mathcal{C}\}$ denote the *intersection closure* of any class $\mathcal{C}$ obtained via intersections of all finite subsets of $\mathcal{C}$'s concepts.

**Lemma 3.2.** *For any concept class $\mathcal{C} \subset \{0,1\}^n$, the intersection closure $\overline{\mathcal{C}}$ is the unique smallest intersection-closed class containing $\mathcal{C}$, and it is contained in any intersection-closed class containing $\mathcal{C}$. Moreover for any $J \subseteq [n]$, we must have $\overline{\mathrm{proj}_J(\mathcal{C})} = \mathrm{proj}_J(\overline{\mathcal{C}})$.*

Next is a useful result on computing the VC dimension of the intersection closure of a class. The key idea is that to verify $\mathrm{proj}_J(\overline{\mathcal{C}})$ shatters, it is necessary and sufficient to show $\overline{\mathcal{C}}$ has $|J| + 1$ vertices which project to the vertices with at most one coordinate $0$.

**Definition 3.3.** Suppose $\mathcal{C}$ is a class in the binary $n$-cube with VC dimension $d$. We will say that $\mathcal{C}$ satisfies the *$k$-close cube condition* if there is a vertex $v \in \{0,1\}^n$ and a complete union of $(n-k-1)$-cubes in the complement of $\mathcal{C}$ so that each cube is distance at most one from $v$. For any $k' > k$, $C$ then also satisfies the $k'$-close cube condition.

**Theorem 3.4.** *Suppose $\mathcal{C}$ is a class in the binary $n$-cube. The smallest VC dimension $\overline{d}$ for the intersection closure of $\mathcal{C}$ over all choices of origin, is the same as the smallest $k$ for which $\mathcal{C}$ satisfies the $k$-close cube condition.*

*Proof.* Suppose first that $\overline{d}$ achieves the smallest VC dimension of the intersection closure of $\mathcal{C}$ over all choices of origin. Let $\overline{C}$ denote the corresponding intersection closure of $\mathcal{C}$. Then any projection $p$ of $\overline{C}$ to a $(\overline{d}+1)$-cube is not onto. Assume $p(\mathcal{C})$ contained the vertex $w = (1,1,\ldots,1)$ and all vertices $w^i = (1,1,\ldots,1,0,1,\ldots,1)$ with $0$ in the $i$th position and all other entries $1$. Then since $p(\overline{C})$ is intersection closed, the image is the whole $(\overline{d}+1)$-cube contrary to assumption. The reason is that taking intersections of combinations of the vertices $w^i$ gives all the vertices in the $(\overline{d}+1)$-cube. So at least one of the vertices $w, w^1, \ldots, w^{\overline{d}+1}$ must not be in $p(\mathcal{C})$. But then there is an $(n-\overline{d}-1)$-cube with anchor of this form in the complement of $\mathcal{C}$ by taking the inverse image under $p$ of this vertex. By choosing $v$ as the vertex with all entries $1$ in the $n$-cube, we have shown that $\overline{C}$ satisfies the $\overline{d}$-close cube condition. For clearly each of our $(n-\overline{d}-1)$-cubes in the complement of $\overline{C}$ has distance at most one from $v$. So this establishes that $\overline{d} \geq k$, where $k$ is the smallest value so that $\mathcal{C}$ satisfies the $k$-close cube condition.

Conversely suppose that $\mathcal{C}$ has a collection of $(n-k-1)$-cubes in its complement, validating that $\mathcal{C}$ satisfies the $k$-close cube condition. We claim that $\overline{d} \leq k'$, where $\overline{d}$ is the VC dimension of the intersection closure of $\mathcal{C}$ for a suitable choice of origin. This shows that the smallest possible VC dimension $\overline{d}$ of the intersection closure of $\mathcal{C}$ over all choices of origin is not larger than the smallest possible $k$ where $\mathcal{C}$ satisfies the $k$-close cube condition.

For suppose that any projection $p$ of $\overline{C}$ to the $(k+1)$-cube is selected. By Lemma 3.2, $p(\overline{\mathcal{C}}) = \overline{p(\mathcal{C})}$. In fact the intersection operator commutes with the projection operator, *i.e.*, $p(I(u,w)) = I(p(u),p(w))$ so the claim follows by Lemma 3.2. But now by assumption, if we choose the origin so that $v$ is the vertex with all entries $1$, then there is an $(n-k-1)$-cube in the complement of $\mathcal{C}$ which has anchor with entries all $1$'s except for at most one entry $0$ and the anchor projects to a vertex of the $(k+1)$-cube. Hence the projection $p$ of $\overline{C}$ is not onto and $\overline{d} \leq k$. $\qquad\square$

We end with a useful property of the intersection closure operator. We will abuse notation by saying a class $A$ is *shortest path closed inside a class $B$*, if $A \subset B$ and any two vertices of $A$ are connected by a shortest path inside $B$. The main idea in the following theorem is that if $\mathcal{C}$ is shortest-path closed, then $X = \{v : v = u \odot w, u, w \in \mathcal{C}\}$ is shortest path closed inside $\overline{\mathcal{C}}$.

**Theorem 3.5.** *If $\mathcal{C}$ is shortest-path closed then its intersection closure $\overline{\mathcal{C}}$ is also shortest-path closed.*

*Proof.* We first observe the easy fact that if $\lambda$ is a shortest path in $\mathcal{C}$ consisting of vertices $v^1, v^2, \ldots, v^k$ and $u \in \mathcal{C}$ then $u \odot \lambda := \{u \odot v^i : i \in [k]\}$ is a shortest path in $\overline{\mathcal{C}}$ possibly with repetitions. For an edge between successive vertices $v^i, v^{i+1}$ is transformed into $u \odot v^i, u \odot v^{i+1}$ which is either an edge of the same colour or a single vertex.

Our main claim that if $\mathcal{C}$ is shortest-path closed then $X = \{v : v = u \odot w, u, w \in \mathcal{C}\}$ is shortest path closed inside $\overline{\mathcal{C}}$. Once this claim is established, the proof follows easily by repeating this step. For we can apply the claim to the class $X$ to prove that the class $\{v : v = u \odot w, u, w \in \mathcal{X}\}$ consisting of all elements of $\mathcal{C}$ together with intersections of up to $4$ elements of $\mathcal{C}$ is shortest-path closed. inside

$\overline{X} = \overline{C}$. Repeating at most $log_2 n$ times, $\overline{C}$ is obtained and is shortest-path closed. (Note that since $v = v \odot v$, elements of $C$ can be written as intersections of pairs of elements of $C$.)

The first step in proving the main claim is to observe that there is a shortest path in $X$ connecting any element $u \in C$ to any element $u \odot v \in X$. To construct this path, start with a shortest path $\lambda$ between $u, v$ in $C$ and then form $u \odot \lambda$. As above this is a shortest path, possibly with repetitions, between $u, u \odot v$ in $X$. We conclude that given two vertices $u, v \in C$, there is a shortest path in $X$ from $u$ to $v$ passing through $u \odot v$, by concatenating shortest paths in $X$ from $u$ to $u \odot v$ and $u \odot v$ to $v$.

To prove the main claim, choose elements $v, v' \in X$. We want to find a shortest path in $\overline{C}$ connecting $v, v'$. Let $v = u \odot w, v' = u' \odot w'$ where $u, w, u', w' \in C$. By the first step, there are shortest paths $\lambda, \lambda'$ in $X$ connecting $u, u'$ and $w, w'$ and passing through $u \odot u', w \odot w'$ respectively.

Next, observe that $v \odot v' = u \odot w \odot u' \odot w' = u \odot u' \odot w \odot w'$. Split $\lambda, \lambda'$ into $\lambda_i, \lambda_i'$ for $i = 1, 2$ so that $\lambda_1, \lambda_1'$ join $u, w$ and $u \odot u', w \odot w'$ respectively and $\lambda_2, \lambda_2'$ join $u \odot u', w \odot w'$ and $u', w'$ respectively. To establish the main claim, it clearly suffices to construct shortest paths in $\overline{C}$ from $v$ to $v \odot v'$ and from $v \odot v'$ to $v'$. By symmetry, we will focus on the first of these.

We will prove a slightly stronger claim. Namely assume $\lambda_1$ consists of ordered vertices $u = z^1, z^2, \ldots, z^k = u \odot u'$ and $\lambda_1'$ has ordered vertices $w = y^1, y^2, \ldots, y^r = w \odot w'$. We will show that a shortest path in $X$ connecting $v, v \odot v'$ can be chosen to have all vertices of the form $z^i \odot y^j$.

To complete the proof, consider the rectangular grid of intersections $z^i \odot y^j$, for $1 \leq i \leq k$ and $1 \leq j \leq r$ with $k, r \geq 1$. Each small square of this grid is either a single vertex, one edge or a 2-cube in $X$ by the previous paragraph. We claim that there is a shortest path between the diagonally opposite entries $v = z^1 \odot y^1, v \odot v' = z^k \odot y^r$ using vertices in this grid.

It suffices to use the boundary of the grid to do this. So for example, consider the first path $z^i \odot y^1$ for $1 \leq i \leq k$, followed by a second path $z^k \odot y^j$ for $1 \leq j \leq r$. By the first observation in this proof, the first path is shortest with possible repetitions, as is the second path. To complete the argument, we need to show that the concatenation of these two paths is also shortest with possible repetitions.

Notice that the path $\lambda_1 = z^1, z^2, \ldots, z^k$ runs between ends $z^1, z^k$ where $z^k$ is between the origin of the binary $n$-cube and $z_1$. So the same follows for the path with possible repetitions $z^i \odot y^1$ for $1 \leq i \leq k$. The second path $z^k \odot y^j$ for $1 \leq j \leq r$ is of the same form and hence the concatenation of these two paths is also shortest with repetitions and the theorem proof is complete. $\qquad\square$

## 4  Extremal intersection-closed classes

In this section, we start with a useful way of finding extremal intersection-closed classes, namely by checking they are shortest-path and intersection-closed. We use this to establish the main result about intersection-closed classes, namely that they can be embedded into extremal intersection closed classes, with the VC dimension $d$ increasing to at most $11d$. We then use a result of Chalopin et al. (2018) to conclude this gives unlabelled compression schemes of size $11d$ for an intersection-closed class of VC dimension $d$.

**Theorem 4.1.** *Suppose that $C$ is shortest-path and intersection closed. Then $C$ is extremal.*

*Proof.* Suppose a projection $p : \{0, 1\}^n \to \{0, 1\}^k$ shatters $C$. We will prove there is a $k$-subcube $c$ of $C$ which witnesses the shattering.

Consider all the vertices of $C$ which map to the vertex $w = (1, 1, \ldots, 1)$ in the $k$-cube. Choose a vertex $v$ in this set which has smallest distance to the origin. (Since $C$ is shortest-path closed and intersection closed, the origin is in $C$ and distance in $C$ is the same as distance in the $n$-cube.)

Our claim is that there are vertices $v^i \in C, i \in [k]$, so that the distance from $v$ to each $v^i$ is one and $p(v^i) = (1, 1, \ldots, 0, \ldots, 1) = w^i$ the vertex in the $k$-cube with all coordinates 1 except a single 0 at the $i$th coordinate. Once this claim is established, it follows that since $C$ is intersection closed, taking the intersections of subcollections of the vertices $\{v^i : i \in [k]\}$ generates the required $k$-subcube $c$ of $C$. Note $c$ witnesses the shattering, since projection commutes with intersection and the vertices $w, w^i$ generate $c$ under the operation of taking intersections.

To prove the claim, choose any vertex $u \in C$ so that $p(u) = w^i$. Since $C$ is intersection closed, $u' = v \odot u \in C$. Then $p(u') = p(v \odot u) = p(v) \odot p(u) = w \odot w^i = w^i$. Note that $u'$ is a vertex

---

**Algorithm 1** Shortest-Path Closure

---
1: **Input:** intersection-closed class $\mathcal{C} \subseteq \{0,1\}^n$
2: $(x_1, \ldots, x_n) \leftarrow$ arbitrary ordering of $[n]$
3: $\mathcal{C}^* \leftarrow \mathcal{C}$
4: **for** $v \in \mathcal{C}$ **do**
5:    **for** $w \in \mathcal{C} \backslash \{v\}, w \le v$ **do**
6:       $\lambda(v,w) \leftarrow \emptyset$     `//v-w shortest path`
7:       $(i,j,v_1) \leftarrow (1,1,v)$
8:       **while** $\|v_i - w\|_1 > 1$ **do**
9:          $v_{i+1} \leftarrow v_i$     `//next point on path`
10:          **while** $v_{i,x_j} = w_{x_j}$ **do** $j \leftarrow j + 1$
11:          $v_{i+1,x_j} \leftarrow 0$     `//step towards w`
12:          $\lambda(v,w) \leftarrow \lambda(v,w) \cup \{v_{i+1}\}$
13:          $i \leftarrow i + 1$
14:       **end while**
15:       $\mathcal{C}^* \leftarrow \mathcal{C}^* \cup \lambda(v,w)$
16:    **end for**
17: **end for**
18: **return** $\mathcal{C}^*$

---

between $v$ and the origin, *i.e.*, for any coordinate where $u'$ has value 1 then $v$ must also. Since $\mathcal{C}$ is shortest-path closed, a shortest path $\lambda$ exists in $\mathcal{C}$ from $v$ to $u'$. Moreover $\lambda$ has vertices in order with distances decreasing to the origin, since $u'$ is between $v$ and the origin.

Let $v'$ be the first vertex on $\lambda$ after $v$. Since $p(\lambda)$ is a shortest path in the $k$-cube from $w$ to $w^i$, it follows that $p(v')$ is either $w$ or $w^i$. If the former, this contradicts our choice of $v$ as a closest vertex in $\mathcal{C}$ to the origin which projects to $w$. If the latter then we have found our vertex $v^i$ with distance 1 to $v$ and which projects to $w^i$. So this completes the proof of the claim and the result. $\qquad\square$

*Remark* 4.2. Chalopin et al. (2018) discusses the significance of properties of extremal intersection-closed classes, also known as *conditional antimatroids*.

*Remark* 4.3. There is a partial order on the binary $n$-cube given by $w \le v$ when $w$ is between $v$ and the origin or equivalently there is a shortest path from $v$ to the origin passing through $w$. Choose an arbitrary ordering on $[n]$. We will denote by $\lambda(v,w)$ the shortest path between $v,w$ with vertices $v = v_1, v_2, \ldots v_k = w$ where $v_i.v_{i+1}$ differ in the first coordinate $x_j$ in the ordering of indices $j$ amongst all coordinates where $v_i, w$ differ.

In the next result, we enlarge an intersection closed class $\mathcal{C}$ to become intersection-closed and shortest-path closed by adding all the vertices in the shortest paths $\lambda(v,w)$ between all pairs $v, w \in \mathcal{C}$.

**Theorem 4.4.** *Any intersection-closed class $\mathcal{C}$ embeds in a shortest-path closed and intersection-closed class $\mathcal{C}^*$ given by Algorithm 1 so that $d^* \le 11d$ where $d, d^*$ are the VC dimensions of $\mathcal{C}, \mathcal{C}^*$ respectively. To construct $\mathcal{C}^*$, for every pair of vertices $v, w \in \mathcal{C}$ with $w < v$, add all the vertices in $\lambda(v,w)$ to $\mathcal{C}$.*

**Lemma 4.5.** *If $\mathcal{C}$ is intersection closed then $\mathcal{C}^*$ as constructed by Theorem 4.4 is intersection closed.*

*Proof.* By definition, for any vertices $u, u' \in \mathcal{C}^*$, there are vertices $v, w, v', w' \in \mathcal{C}$ so that $u \in \lambda(v,w), u' \in \lambda(v',w')$. Moreover, $w \le v, w' \le v'$.

Since $\mathcal{C}$ is intersection closed, both $v \odot v'$ and $w \odot w'$ are in $\mathcal{C}$. We claim that $u \odot u'$ is on the path $\lambda(v \odot v', w \odot w')$—note that as part of this claim, we require that $w \odot w' \le v \odot v'$.

We begin with the latter claim. This is easy—$w \odot w'$ has entry 1 at precisely the coordinates where both $w, w'$ have entry 1. But since $w \le v, w' \le v', v, v'$ has entry 1 at all the coordinates where $w, w'$ respectively has entry 1. So for any coordinate where $w \odot w'$ has entry 1, it follows immediately that both $v, v'$ have entry 1 and so does $v \odot v'$.

Next, the set of coordinates where $v$ has entry 1 and $u$ has entry 0 forms an initial segment of the coordinates where $v$ has entries 1, using the ordering of the coordinates, since $u \in \lambda(v,w)$. A similar analysis applies to the set of coordinates of $u'$ which are 0 but the entries for $v'$ are 1.

To complete the proof that $\mathcal{C}^*$ is intersection closed, we show that the set of coordinates where the entries of $v \odot v'$ are 1 and the entries of $u \odot u'$ are 0, forms an initial segment of the ordering of the coordinates where the entries of $v \odot v'$ are 1. It will then follow that $u \odot u' \in \lambda(v \odot v', w \odot w')$.

Now the set of coordinates where $v \odot v'$ has entries 1 are exactly those coordinates where both $v, v'$ have entries 1. Moreover $u \odot u'$ has entry 0 at a coordinate if either $u$ or $u'$ has entry 0. We complete the proof using an argument by contradiction. Suppose that there is a coordinate $x_i$ such that $u \odot u'$ has entry 1 but this occurs before a coordinate $x_j$ where $u \odot u'$ has entry 0 and moreover $v \odot v'$ has entry 1. Then at $x_j$, either $u$ or $u'$ has entry 0. Without loss of generality assume the former. Since $u \odot u'$ has entry 1 at $x_i$ it follows that $u$ has entry 1 at $x_i$. Note also that $v$ has entry 1 at both $x_i, x_j$. But this is a contradiction since the coordinates where $v$ has entry 1 but $u$ has entry 0 should form an initial segment of the set of coordinates where $v$ has entry 1. But we have found $j > i$ where $u$ has entry 1 at $x_i$ and 0 at $x_j$. This establishes that $\mathcal{C}^*$ is intersection closed. $\qquad\square$

**Lemma 4.6.** *If $\mathcal{C}$ is intersection closed then $\mathcal{C}^*$ as constructed by Theorem 4.4 is shortest-path closed.*

*Proof.* By Lemma 4.5, it suffices to show that if $v, w \in \mathcal{C}^*$ then there are shortest paths from $v, w$ to $v \odot w$, since putting these together will give a shortest path from $v$ to $w$. Next, it suffices to show there is a shortest path in $\mathcal{C}^*$ between any pair of vertices $v, u$ in $\mathcal{C}^*$, where $u \leq v$, for we can then apply this to the pairs $v, v \odot w$ and $w, v \odot w$.

We claim that there is a vertex $v_1 \in \mathcal{C}$ so that $u \leq v_1, v \leq v_1$ and there are shortest paths in $\mathcal{C}^*$ joining both $v_1, u$ and $v_1, v$. Firstly, since $v \in \mathcal{C}^*$, there are vertices $v_1, v_2 \in \mathcal{C}$ so that $v$ is in $\lambda(v_1, v_2)$. Hence $v \leq v_1$ and there is a shortest path in $\mathcal{C}^*$ from $v_1$ to $v$, namely a segment of $\lambda(v_1, v_2)$.

Next, since $u \leq v$, then $u \leq v_1$. Moreover $u$ lies on a path $\lambda(u_1, u_2)$ where $u_1, u_2 \in \mathcal{C}$, since $u \in \mathcal{C}^*$. Then $v_1 \odot u_1 \in \mathcal{C}$ and $v_1 \odot u_1 \leq u_1$. Since $u \leq v_1, u \leq u_1$, it follows that $u \leq v_1 \odot u_1$.

Notice that $u$ must lie on $\lambda(v_1 \odot u_1, u_2)$. To verify this, observe all the coordinates where entries of $v_1 \odot u_1$ differ from entries of $u$ are also coordinates where $u_1$ entries differ from $u$ entries. Consequently, coordinates where entries of $u$ and $u_2$ differ, are lower in the coordinate ordering than those where entries of $v_1 \odot u_1$ and $u$ differ. So this confirms that $u \in \lambda(v_1 \odot u_1, u_2)$.

Combining $\lambda(v_1, v_1 \odot u_1)$ and $\lambda(v_1 \odot u_1, u)$ gives a shortest path from $v_1$ to $u$ in $\mathcal{C}^*$. So this completes the proof of the claim about $v_1$.

To recapitulate, we have constructed shortest paths in $\mathcal{C}$ from $v_1$ to both $u, v$ in $\mathcal{C}^*$, where $v \leq v_1, u \leq v_1, u \leq v$. To complete the proof that there is a shortest path from $v$ to $u$ in $\mathcal{C}$, we use induction on the Hamming distance from $v_1$ to $v$. To begin the induction, suppose this distance is 0. Then clearly we are done, since there is a shortest path in $\mathcal{C}^*$ from $v$ to $u$.

Assume the distance from $v_1$ to $v$ is $k$ and the result is true for $v_1*, v*, u*$ as above, where the distance from $v_1*$ to $v*$ is less than $k$. Let $v'$ be the first vertex in the shortest path from $v_1$ to $v$. Our aim is to show that there is a shortest path in $\mathcal{C}^*$ from $v'$ to $u$. By induction, it then follows there is a shortest path in $\mathcal{C}^*$ from $v$ to $u$.

The argument is similar to that in Theorem 3.5. Let $v_1 = y^1, y^2, \ldots, y^r = u$ be a shortest path in $\mathcal{C}^*$. We can construct a shortest path $v' = v' \odot y^1, v' \odot y^2, \ldots, v' \odot y^r = u$ where there is a single repetition. Namely, since one of the edges joining $y^j, y^{j+1}$ must have the same colour as the edge between $v_1, v'$, then $v' \odot y^j = v' \odot y^{j+1}$. All other pairs of vertices are distance one apart and since $\mathcal{C}^*$ is intersection closed, this shortest path is in $\mathcal{C}^*$. So this completes the proof. $\qquad\square$

*Proof.* (Theorem 4.4) It remains to prove that the VC dimension of $\mathcal{C}^*$ is at most $11d$.

Choose any projection $p : \{0, 1\}^n \to \{0, 1\}^{11d}$. By Sauer's Lemma, since the VC dimension of $p(\mathcal{C})$ is at most $d$, the cardinality $|p(\mathcal{C})|$ is at most $F(d; 11d, \frac{1}{2}) \times 2^{11d}$ where $F(k; n, p)$ is the cumulative binomial probability function.

Algorithm 1 constructing $\mathcal{C}^*$ involves finding and adjoining shortest paths connecting suitable ordered pairs of points in $\mathcal{C}$. So $|\mathcal{C}^*| \leq n \times |\mathcal{C}|^2$ . Similarly we see that $|p(\mathcal{C}^*)| \leq 11d \times |p(\mathcal{C})|^2$, since the process of adjoining shortest paths is preserved under projection. Hence $|p(\mathcal{C}^*| \leq 11d \times F(d; 11d, \frac{1}{2})^2 \times 2^{22d} \times \frac{1}{2}$. Note that as pairs $(v, w)$ are ordered by requiring that $w \leq v$, we get a factor of $\frac{1}{2}$. We claim that this upper bound is strictly less than $2^{11d}$. Then $p$ cannot shatter and the VC dimension of $\mathcal{C}^*$ is at most $11d - 1$.

The key idea is to use Chernoff's inequality to estimate $F(d; 11d, \frac{1}{2})$. This is:

$$F\left(d; 11d, \frac{1}{2}\right) \leq \exp\left((-11d) \times D\left(\frac{1}{11}\bigg\|\frac{1}{2}\right)\right) ,$$

where $D(\frac{1}{11}\|\frac{1}{2}) = \frac{1}{11}\log\frac{2}{11} + (1 - \frac{1}{11})\log 2(1 - \frac{1}{11})$. Substituting this into the previous upper bound and simplifying, we obtain $|p(\mathcal{C}^*)| \leq \frac{11d \times 2^{22d}}{2}\exp(22d\log 11 - 2d\log 2 - 20d\log 20)$. We can further simplify this to $|p(\mathcal{C}^*)| \leq 11d \times 11^{22d} \times 10^{-20d} \times \frac{1}{2}$. To complete the argument, we need to show the right side is strictly less than $2^{11d}$. Hence it suffices to show that

$$(11d)^{\frac{1}{d}} \times 11^{22} \times 10^{-20} \times \frac{1}{2^{\frac{1}{d}}} < 2^{11}$$

Now if $d \geq 5$ then $11^{\frac{1}{d}} < 1.7$ and $d^{\frac{1}{d}} < 1.4$. So we can compute the left side is at most $1.7 \times 1.4 \times 121 \times \frac{11}{10}^{20} < 1937$ whereas the right side is 2048. (Here we leave out the factor $2^{-\frac{1}{d}}$.)

If $d = 4$ then $\frac{11}{2}^{\frac{1}{4}} < 1.6$ and $4^{\frac{1}{4}} < 1.42$. So we can compute the left side is as at most $1.6 \times 1.42 \times 121 \times \frac{11}{10}^{20} = 1810$ whereas the right side is 2048. So this completes the case when $d \geq 4$. For the other cases, it is convenient to work directly, as Chernoff's inequality is not sharp.

For $d = 1$, $|p(\mathcal{C})| \leq 12$, as it has VC dimension at most 1 in the 11-cube. Hence $|p(\mathcal{C}^*)| \leq 11 \times 12^2 \times \frac{1}{2} = 792$ which is less than 2048.

Next suppose $d = 2$. $|p(\mathcal{C})| \leq 254$, as it has VC dimension at most 2 in the 22-cube. Hence $|p(\mathcal{C}^*)|$ is bounded above by $22 \times 254^2 \times \frac{1}{2} = 709,676$ which is less than $2048^2 = 4,194,304$.

Finally assume $d = 3$. $|p(\mathcal{C})| \leq 6018$, as it has VC dimension at most 3 in the 33-cube. Hence $|p(\mathcal{C}^*)|$ is bounded above by $33 \times 6018^2 \times \frac{1}{2} = 597,569,346$ which is less than $2048^3 = 8,589,934592$. $\square$

**Corollary 4.7.** *If $\mathcal{C}$ is an intersection-closed class, then $\mathcal{C}$ has a (corner peeling) unlabelled compression scheme of size at most $11d$.*

*Proof.* This follows immediately by combining Theorem 4.4 and (Chalopin et al., 2018, Theorem 4.9). $\square$

# 5 Unlabelled compression schemes for extremal classes

The aim is to give a condition on extremal classes which implies that they have unlabelled compression schemes. We then give some situations where this criterion is satisfied. We conjecture that it is always valid.

For a maximum class $\mathcal{C}$ of VC dimension $d$, a reduction is a maximum subclass of VC dimension $d - 1$. So a reduction contains cubes with the same set of colours as any non-maximal cube of $\mathcal{C}$. It will turn out this is the key property we require of extremal classes to have unlabelled compression schemes of size $d$, namely they have proper extremal subclasses which have the same properties as reductions of maximum classes.

## 5.1 High-level strategy

The idea is inspired by Chalopin et al. (2018). Start with a pair $(\mathcal{C}, \mathcal{D})$ consisting of an extremal class and a proper extremal subclass. The condition is that for every cube in $\mathcal{C}$ which is not maximal, there is a cube with the same colours in $\mathcal{D}$. We call this the *cubical colour condition* of the pair $\mathcal{C}, \mathcal{D}$ and abbreviate it by *ccc*.

We will construct a bijection $\phi$ between the set $\mathbb{S}$ of maximal cubes $c$ of $\mathcal{C}$ not in $\mathcal{D}$ and $\mathcal{C} \setminus \mathcal{D}$, so that $v = \phi(c) \in c$. Note the cardinalities of these two sets are the same, by the Sandwich Lemma 2.5 applied to the two extremal classes $\mathcal{C}, \mathcal{D}$. Then $\phi^{-1} : \mathcal{C} \setminus \mathcal{D} \to \mathbb{S}$ gives a representation map for $\mathcal{C} \setminus \mathcal{D}$, which also separates the vertices of $\mathcal{C} \setminus \mathcal{D}$ from the vertices of $\mathcal{D}$. Therefore, any representation of $\mathcal{D}$ can be combined with the representation of $\mathcal{C} \setminus \mathcal{D}$ to give a representation of $\mathcal{C}$.

Suppose we can find a sequence of proper pairs of extremal subclasses $(\mathcal{C}, \mathcal{D}), (\mathcal{D}, \mathcal{E}), \ldots, (\mathcal{X}, \mathcal{Y})$, all satisfying *ccc*, ending with VC $(\mathcal{Y}) \leq 2$. Then we can start with a representation map of the VC-2

class $\mathcal{Y}$ (Chalopin et al., 2018) and extend to a representation of $\mathcal{X}$, then extending the representation all the way to $\mathcal{C}$ using this strategy.

## 5.2 Detailed construction

We construct $\phi : \mathcal{C} \setminus \mathcal{D} \to \mathbb{S}$ and show it is a bijection. Firstly, denote the extremal class which is the complement of $\mathcal{D}$ by $\mathcal{D}^*$. Then every vertex of $\mathcal{C} \setminus \mathcal{D}$ is in at least one maximal cube of $\mathcal{D}^*$.

Define two cubes $c, c^*$ in $\{0,1\}^n$ to be *complementary* if the dimensions $d, d^*$ of $c, c^*$ satisfy $d + d^* = n$ and $c \cap c^*$ is a single vertex. Recall that $\mathbb{S}$ is the set of maximal cubes $c$ of $\mathcal{C}$ not in $\mathcal{D}$. Denote the set of maximal cubes of $D^*$ which intersect $\mathcal{C}$ by $\mathbb{S}^*$.

**Lemma 5.1.** *Assume $(\mathcal{C}, \mathcal{D})$ is a proper pair of extremal classes satisfying ccc. If $c \in \mathbb{S}$, then there is a complementary maximal cube $c^*$ of $\mathcal{D}^*$. Conversely, if $c^*$ is a maximal cube of $\mathbb{S}^*$, then there is a complementary cube $c \in \mathbb{S}$. A maximal cube $c^*$ of $\mathbb{S}^*$, cannot have two different complementary cubes in $\mathbb{S}$.*

Given Lemma 5.1, we will call $v = c \cap c^*$ the *special vertex* of $c$. We now note properties of $v$, assuming Lemma 5.1.

**Corollary 5.2.** *Assume that $(\mathcal{C}, \mathcal{D})$ is a proper pair of extremal classes satisfying ccc. Let $v = c \cap c^*$ where $c \in \mathbb{S}$ and $c^* \in \mathbb{S}^*$ are complementary maximal cubes. Then*

- *The set of vertices $v = c \cap c^*$, for all pairs of complementary maximal cubes $c \in \mathbb{S}$ and $c^* \in \mathbb{S}^*$, is $\mathcal{C} \setminus \mathcal{D}$.*
- *The mappings between $c, c^*, v$ are bijections between $\mathbb{S}, \mathbb{S}^*$ and $\mathcal{C} \setminus \mathcal{D}$.*

*Proof.* Lemma 5.1 gives a bijection between $\mathbb{S}$ and $\mathbb{S}^*$, by the mapping from cubes $c$ to complementary cubes $c^*$. Moreover by the Sandwich Lemma applied to extremal classes $\mathcal{C}, \mathcal{D}$, the cardinalities of $\mathcal{C} \setminus \mathcal{D}, \mathbb{S}^*$ are the same. Since the map from maximal cubes in $\mathbb{S}$ to special vertices is an injection by Lemma 5.1, it must be a bijection onto $\mathcal{C} \setminus \mathcal{D}$. $\qquad\square$

*Proof of Lemma 5.1.* Let $p : \{0,1\}^n \to \{0,1\}^d$ be the projection to the colours of $c \in \mathbb{S}$. Recall $ccc$ for the pair $\mathcal{C}, \mathcal{D}$ says that for any cube of $\mathcal{C}$ which is not maximal, there is a cube of $\mathcal{D}$ with the same set of colours. So $p(\mathcal{D}) = \{0,1\}^d \setminus \{w\}$ for some vertex $w$. For the only other case is that $p(\mathcal{D}) = \{0,1\}^d$. But since $\mathcal{D}$ is extremal, this would imply $\mathcal{D}$ contained a cube with the same colours as $c$. This contradicts the assumption that $c$ is maximal in $\mathcal{C}$, since for an extremal class, maximal cubes are unique for their sets of colours.

Define $v, c^*$ by $v = p^{-1}(w) \cap c, c^* = p^{-1}(w)$ respectively. Clearly $c \cap c^* = v$. It remains to show that $c^*$ is maximal in $\mathcal{D}^*$ and $c^*$ cannot intersect some other cube of $\mathbb{S}$ in a single vertex.

Suppose that $c^*$ was properly contained in a larger cube $\tilde{c}$ in $\mathcal{D}^*$. Then $p(\tilde{c})$ would properly contain $\{w\}$. Moreover $\tilde{c} = p^{-1}p(\tilde{c})$. But this contradicts our observation above that $p(\mathcal{D}) = \{0, 1|^d \setminus \{w\}$. So this completes the argument that $c^*$ is maximal.

Next, assume that $c^*$ is complementary to a second maximal cube $\hat{c} \in \mathbb{S}$. Consider the projection $\hat{p}$ of $\{0,1\}^n$ to $\{0,1\}^{\hat{d}}$ given by the $\hat{d}$ colours of $\hat{c}$. As above, $\hat{p}(\mathcal{D})$ consists of $\{0,1\}^{\hat{d}}$ with a single vertex $\hat{w}$ removed. Moreover, $\hat{p}^{-1}(\hat{w})$ is a maximal cube $c'$ of $\mathcal{D}^*$.

Now as $c^* \cap \hat{c} = u^*$ is a single vertex, $\hat{p}(c^*) = \hat{p}(u^*) = u$. As the cubes $c^*, \hat{c}$ are complementary, $c^* = \hat{p}^{-1}(u)$. But now we see that the colours of the anchor of $c^*$ are both the colours of $c$ and of $\hat{c}$. This gives a contradiction, since two maximal cubes of $\mathcal{C}$ cannot have the same set of colours. This completes the proof of Lemma 5.1. $\qquad\square$

Finally we show that the map from a special vertex $v$ to the colours of the corresponding maximal cube $c$ of $\mathbb{S}$ gives a representation mapping of $\mathcal{C} \setminus \mathcal{D}$ which also separates $\mathcal{C} \setminus \mathcal{D}$ from $\mathcal{D}$.

The second property follows immediately. For recall that if $p$ is the projection given by the $d$ colours of $c$, then $p(\mathcal{D}) = \{0,1\}^d \setminus \{p(v)\}$. So the colours of $c$ clearly separate $v$ from all the vertices of $\mathcal{D}$. By this we mean that given any vertex $w \in D$, $v, w$ differ on at least one coordinate from the colours of $c$.

Next suppose two vertices $v, v'$ have the same values at all the coordinates corresponding to the colours of their corresponding cubes $c, c' \in \mathbb{S}$ of dimensions $d, d'$ respectively. Project $\{0,1\}^n$ to the union of the colours of $c, c'$ by a mapping $p''$. The image of $p''$ is a cube $\{0,1\}^{d''}$ of dimension $d'' \leq d + d'$. In this cube, $\mathcal{C}, \mathcal{D}$ map to extremal classes $p''(\mathcal{C}) = \mathcal{C}'', p''(\mathcal{D}) = \mathcal{D}''$. We want to check that the pair of extremal classes $(\mathcal{C}'', \mathcal{D}'')$ satisfy $ccc$. But this follows easily. Choose any non maximal cube $\tilde{c} \in \mathcal{C}''$. As $\mathcal{C}$ is extremal, there is a cube $c_0$ in $\mathcal{C}$ which projects one-to-one to $\tilde{c}$ and therefore has the same colours as $\tilde{c}$. Since $c_0$ is not maximal, there is a cube $c_1$ in $\mathcal{D}$ with the same colours as $c_0$. Then $p''(c_1)$ is the required cube in $\mathcal{D}''$ with the same colours as $\tilde{c}$ as required.

Finally, $c, c'$ project one-to-one by $p''$ to cubes with colours spanning all of $\{0,1\}^{d''}$. We claim that $p''(v) \neq p''(v')$. Since $p''(v), p''(v')$ are the special points for $p''(c), p''(c')$ in $\{0,1\}^{d''}$, they cannot coincide, by Lemma 5.1. But then $p''(v), p''(v')$ must have at least one different value of their coordinates at the colours of $c, c'$ and hence the same is true for $v, v'$. So this gives a representation for $\mathcal{C} \setminus \mathcal{D}$, as we have verified the no-clashing condition.

## 6 Applications

We apply the construction in Section 5 to two special collections of extremal classes.

**Theorem 6.1.** *Suppose that $\mathcal{C}$ is an extremal class of VC dimension $d$ containing a maximum class $\mathcal{D}$ of VC dimension $d - 1$. Then $\mathcal{C}$ has a $d$-size unlabelled compression scheme.*

*Proof.* This follows immediately by the construction in Section 5. We only have to check that there is a sequence of pairs of extremal classes starting at $\mathcal{C}, \mathcal{D}$, all satisfying $ccc$. But this is easy since any maximum class of VC dimension $i$ is a union of a complete collection of maximal cubes of dimension $i$. Moreover any maximum class of VC dimension $i$ has reductions which are products of maximum classes of VC dimension $i - 1$ with $\{0, 1\}$. $\qquad\square$

**Theorem 6.2.** *Suppose that $\mathcal{C}$ is an extremal class of VC dimension 2. Then $\mathcal{C}$ contains a proper extremal class $\mathcal{D}$ so that for every edge of $\mathcal{C}$, there is an edge of $\mathcal{D}$ of the same colour. Hence $\mathcal{C}$ has a compression scheme of size 2.*

*Proof.* Rubinstein and Rubinstein (2008) develops the idea of splitting classes along a reduction (associated to a coordinate projection). Choose a reduction $\mathcal{C}^1$ of $\mathcal{C}$ which splits off a component $\mathcal{E}$ with a smallest number of vertices, amongst all choices of reductions and complementary components. Let $\mathcal{D} = \mathcal{C} \setminus \mathcal{E}$. We claim that $\mathcal{D}$ is extremal and contains edges of each colour in $\mathcal{C}$.

The reason is that for any other reduction $\mathcal{C}^2$ of $C$, if $\mathcal{C}^2 \cap \mathcal{E} \neq \emptyset$, then $C^2 \cap \mathcal{C}^1 \neq \emptyset$. For if $C^2 \cap \mathcal{C}^1 = \emptyset$, there is a smaller component which $C^2$ splits off from $\mathcal{E}$, which is a contradiction. But $C^2 \cap \mathcal{C}^1 \neq \emptyset$ implies that $\mathcal{D}$ has edges of each colour of $\mathcal{C}$. Showing $\mathcal{D}$ is extremal is straightforward. $\qquad\square$

*Remark* 6.3. Chalopin et al. (2018) showed that VC-2 extremal classes have unlabelled 2-compression schemes. Theorem 6.2 gives an alternate proof using our criterion for extremal classes to have such schemes in Section 5.

## 7 Conclusion

Chalopin et al. (2018) constructed unlabelled compression schemes for maximum classes and extremal intersection-closed classes, while Moran and Warmuth (2016) developed labelled compression schemes for extremal classes. Here we have shown that intersection-closed classes can be embedded into extremal intersection-closed classes with an increase in VC dimension from $d$ to $11d$ and so have $11d$-size unlabelled compression schemes. We simplify and extend the $d$-size unlabelled compression schemes for maximum classes due to Chalopin et al. (2018), to apply to extremal classes of VC dimension $d$ which contain maximum classes of VC dimension $d - 1$. We also give a sufficient condition for extremal classes to admit such compression schemes.

## Acknowledgments and Disclosure of Funding

We acknowledge support from the Australian Research Council Discovery Project DP220102269.

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
