# OpenReview forum: "Unlabelled Sample Compression Schemes for Intersection-Closed Classes and Extremal Classes"
_NeurIPS.cc/2022/Conference — NeurIPS 2022 Accept_

### Official Review · Reviewer_L9jW · 2022-07-04

**Rating:** 7
**Confidence:** 3
**Soundness:** 3 good
**Presentation:** 3 good
**Contribution:** 3 good

**Summary:**

This paper studies the classic problem of whether concept classes with VC dimension d admit compression sizes of, say O(d) or even poly(d). In this paper, the authors show that extremal classes that satisfy a particular property admit unlabelled compression schemes of size d. They give three applications of this. First, they show that extremal classes of VC dimension d which contain maximal classes of dimension d-1 admit unlabelled compression schemes of size d. Second, they show that if extremal classes of VC dimension 2 have unlabelled compression schemes of size 2. Finally, they show that intersection-closed classes with VC dimension d have unlabelled compression schemes of size O(d).

**Questions:**

- Line 31, could not understand what this sentence is saying.
- Line 31, what are VC-d classes? Please define.
- Line 57, I found this paragraph somewhat hard to parse. It says that we can embed intersection-closed classes into extremal intersection-closed classes with a linear increase in VC-dimension. But then later, it is stated that intersection-closed classes have unlabelled compression linear in their VC-dimension. The latter would suggest there is no blow-up? From Theorem 4.4, I think you should say that we can embed intersection-closed classes into extremal intersection-closed classes with only a constant blow-up in VC-dimension.


**Limitations:**

Yes

**Strengths And Weaknesses:**

**Strengths.** This is a fundamental problem in learning theory and the authors make some progress in this area. The paper is also reasonably well-written for the most part. The results in this paper may be of interest to other researchers who are also interested in better understanding compression.

**Weaknesses.** The main drawback is that the contribution is somewhat incremental. Nonetheless, it is still progress in this area so it may be worth presenting to the ML theory community.

---

> ### Author Response · Authors · 2022-08-02
> **Author Response to Official Review**
>
> We thank the reviewer for their feedback. We first respond to the discussion of significance.
>
> Over the first 30 years since Littlestone & Warmuth (1986) conjectured sample compressibility, all major results on compression were proved for maximum classes (those concept classes with the largest cardinality possible given their VC dimension). Such concept classes have special structure that doesn’t appear to shed light on the general conjecture. Despite numerous papers on compression by a number of learning theorists in the intervening years, it was only much later (Moran & Warmuth 2016) when the first results for other major families of classes began to appear. More progress has been made forming connections outside compression: There have been applications to teaching (*e.g.*, teaching dimension), to generalisation error bounds, and to adaptive data analysis most recently. Positive results on compressibility for new families of concept classes have proven to be exceedingly difficult.
>
> Showing that maximum classes have size $O(d)$ unlabelled compression schemes had remained elusive for some time, having been widely studied. Kuzmin & Warmuth (2007) presented a construction they asserted to work (their tail matching scheme), alongside several new open problems stimulating further research (our references cover some of this work). Their paper was particularly influential with many spinoffs by other researchers building on their results.
>
> Unfortunately Kuzmin & Warmuth's proof has been shown not only to be wrong, but has a counterexample (Chalopin *et al.* 2018). This has meant all the spinoffs are wrong also. Chalopin *et al.* (2018) give the first correct proof, but it is very long and difficult to gain insight into why it works. Understanding 'why' is critical to adapting new approaches to more general families. Our proof is substantially shorter and highlights the key property of maximum classes required for the scheme to work. We are then able to extend the approach to the important context of extremal classes - a strict generalisation of maximum classes - with an interesting structural assumption about extremal classes. In the paper we give useful instances where this assumption is valid. Our results contribute new types of extremal classes satisfying the compression conjecture.
>
> Our other main contribution is to show that all intersection-closed classes satisfy the conjecture. (These classes arise in discrete and computational geometry.) We achieve this by embedding such classes into extremal intersection-closed classes, where the VC dimension increases by at most a factor of 11.
>
> ### Questions
> Responses to detailed questions below. Note that we have submitted a **revised paper, and a supplementary file colour coding the diff**.
> * *Line 31 unclear sentence:* We have reworded the introductory text on the conjecture’s history in our revised submission (remains at line 31).
> * *Line 31 VC-$d$ classes:* “VC-$d$” is short-hand for having VC dimension $d$. We have clarified this on lines 25-26 in the revision.
> * *Line 57 linear increase to VC-dim:* The two statements referred to are not inconsistent. We have reworded the first statement now lines 58-59 in the revision. When embedding an intersection-closed class of VC-dim $d$ into an extremal intersection-closed class, the VC-dim may increase to at most $11d$. Additionally, since the latter type of class can be compressed with size equal to its VC-dim, it follows that the latter class can be compressed to size $11d$. Finally, since compression schemes can be restricted to subclasses (without losing the scheme’s size), the consequence is that the original intersection-closed class can be compressed to size $11d$.

---

> > ### Comment · Reviewer_L9jW · 2022-08-08
> > **response**
> >
> > Thanks to the authors for the detailed discussion on some of the history about this problem. In light of that, I have adjusted to my score to a 7. Thanks also for addressing my questions.

---

> > > ### Author Response · Authors · 2022-08-09
> > > **Author reply**
> > >
> > > Thank you Reviewer L9jW for engaging with us during discussion period and for your support of the paper.

---

### Official Review · Reviewer_Ara5 · 2022-07-09

**Rating:** 5
**Confidence:** 3
**Soundness:** 3 good
**Presentation:** 2 fair
**Contribution:** 2 fair

**Summary:**

This paper studies compression schemes which is a formalism that is aimed at classifying learnability via compressability.
It is conjectured in the literature that VC dimension d should imply the concept class should have compression scheme of size at most d.  The main result of the paper is an 11d upper bound on the size of a compression scheme of concept classes that are closed under intersection and have VC dimension d.

**Questions:**

What is the relevance of your results for the NeurIPS crowd? Are there any takeaways or  insights for people who have not encountered or studied compression schemes?

**Limitations:**

The narrow scope and results obtained in this paper.

**Strengths And Weaknesses:**

The paper makes some progress on the relation between compression schemes and VC dimension: a well studied subject in
computational learning theory.
The results are highly specific and seem to be relevant for a very small crowd.
I think this paper should be a better fit for a smaller theory conference such as COLT or ICALP.
Furthermore, the presentation of the paper is not ideal with several awkwardly phrased sentences and other issues
(e.g., the end of 33, 55 and using undefined term of VC-d in 33).

---

> ### Author Response · Authors · 2022-08-02
> **Author Response to Official Review**
>
> We thank the reviewer for their assessment of the paper.
>
> The review’s primary concern is relevance with NeurIPS. Most learning theorists engage with NeurIPS, which includes “Theory (e.g., control theory, learning theory, algorithmic game theory)” among 11 overall topics in its 2022 call for papers. NeurIPS regularly publishes some of the best learning theory work. And like many other topics covered in NeurIPS outside learning theory, many such papers appeal to smaller groups within the broader ML community. The appeal of NeurIPS is its broadest possible scope.
>
> Compression has deep connections to adaptive data analysis (the task of generalising to a population, when reusing a sample repeatedly - a task that is very relevant to practical deployments - see the ladder mechanism by Blum & Hardt (ICML2015) for a practical example. Sample compression has been used for example to derive generalisation bounds for SVMs - cited in our paper. It has also been connected to the teaching dimension, which is related to active learning. While these are important connections, most research in sample compression is fundamental: it sheds light on the geometry of VC classes which are clearly themselves fundamental to learning.
>
> ### Questions
> Responses to detailed questions below. Note that we have submitted a **revised paper, and a supplementary file colour coding the diff**.
> * *Lines 33 and 55 awkward wording:* We have reworded these passages in the revised paper (lines 33 and 55 respectively).
> * *Line 33 undefined term “VC-$d$”*: We have defined this term in line 26 of the revised paper where it is first used: “those [concept classes] having VC dimension $d$”.
>
> We’d be happy to make further edits to the paper during the author-reviewer discussion if the reviewer has any other concerns.

---

### Official Review · Reviewer_XGVd · 2022-07-11

**Rating:** 6
**Confidence:** 2
**Soundness:** 3 good
**Presentation:** 4 excellent
**Contribution:** 3 good

**Summary:**

This paper studies sample compression schemes for extremal concept classes. The authors show that a criterion (containing a maximum class of smaller VC dimension by 1) for an extremal class C to admit a compression scheme of size $VC(C)$.
The authors also show that intersection-closed concept classes have compression schemes linear in their VC dimension.


**Questions:**

### Comments/suggestions for accessibility and clarity

- Shattering should be included in Definition 2.1 in relation to the definition of projection, as it is not introduced anywhere
- Extremal classes could have their own definition where “which maps bijectively to the $|I|$-cube…” is accompanied by an equation; it would be helpful to give examples of classes that are extremal but not maximal.
- (Unlabeled) sample compression schemes are nowhere formally defined in the paper despite being the central object of study
- Section 4 does not have any text apart from the theorem statements and their proofs (and Sections 3, 5 and 6 —apart from 5.1— also lack text). I think some text should be included to introduce the results, their significance, and the main idea behind the proof, perhaps at the cost of moving some full proofs to the appendix and give a proof sketch if the page limit is an issue.



**Limitations:**

Yes and N/A


**Strengths And Weaknesses:**

### Orginality and Quality

The authors make progress towards understanding conditions under which concept classes have unlabeled compression scheme of size equal to their VC dimension. Their results and proofs seem sound and the proof techniques are interesting.

### Significance

This is a bit outside of my are of expertise, but, given the related work mentioned in the paper, the results seem of interest to the learning theory community.

### Presentation and Clarity

Despite the interesting subject matter, I found the main part of the paper quite unaccessible and lacking in its presentation, especially considering that the audience for NeurIPS is quite broad (this is a shame, as the introduction section is itself enjoyable to read and seems quite thorough). For example, the main sections of the paper (except 5.1) barely contain any text contextualizing the results or giving a high-level idea of the proof techniques. While terse papers can be suitable for theory-heavy conferences, I believe that the presentation is not adequate for NeurIPS.

I am open to changing my score (5 -> 6/7) should the authors commit to make the paper more accessible by addressing the comments below.

---

> ### Author Response · Authors · 2022-08-02
> **Author Response to Official Review**
>
> We thank the reviewer for their constructive feedback and supportive assessment of the quality of the contributions and interest to the learning theory community. The reviewer has offered a list of comments to which we commit to address. We have submitted a revised paper incorporating all of the reviewer’s suggestions for improving accessibility, and have submitted a supplemental file with a colour-coded diff for this revision.
>
> We have added contextualising material at the beginning of Sections 4,5,6, and within these sections, to make the paper more accessible to a general audience. We agree that this contextualisation is important for NeurIPS and greatly appreciate the reviewer’s suggestions.
>
> ### Questions
> Responses to detailed questions follow.
> * *Definition 2.1 should include shattering:* We have included this as suggested (now line 74 in the revised paper).
> * *Definition of extremal classes and examples:* As suggested we have inserted new Definition 2.4 including both text (the bijection condition) and a simple mathematical expression. Following this (now lines 94-96 in the revised paper) we offer examples of VC-dimension 1 classes that are extremal but not maximum.
> * *Definition of unlabelled compression schemes:* We have reworked the text around representations and unlabelled compression schemes as suggested. Lines 105-109 provide intuition while Definition 2.6 contains the formal definition of unlabelled compression schemes. These are the same thing as representation mappings but by a different name - apologies that this was ambiguous initially.
> * *Add text to introduce and contextualise results:* This has now been done throughout the revised paper, thankyou for encouraging us to do this. In particular we have added text: at the starts of Sections 4-6 to introduce the results and their significance. We have attempted to summarise the main ideas of key proofs for Theorems 3.4, 3.5, 4.4. In lines 303-307 of the revision, for our Section 5 simplification of (Chalopin *et al.* 2018), we discuss our crucial idea that if extremal classes have subclasses which are like reductions of maximum classes, then they have $d$-unlabelled compression schemes. To accommodate these additions we have moved the proof of Theorem 3.4 to a new Appendix A found after the references and NeurIPS checklist.
>
> We’d be happy to make further edits to the paper during the author-reviewer discussion if any issues persist or new concerns arise.

---

> > ### Comment · Reviewer_XGVd · 2022-08-03
> > **Revised version**
> >
> > Thank you for the revised version and for taking my and other reviewers' comments into account. I think the paper is much more readable and accessible now, and have increased my score (Presentation: 2 -> 4; Rating: 5 -> 6) to reflect this.

---

> > > ### Author Response · Authors · 2022-08-05
> > > **Author reply**
> > >
> > > Thank you Reviewer XGVd for engaging with us during discussion period and for your support of the paper.

---

### Official Review · Reviewer_iWvH · 2022-07-12

**Rating:** 7
**Confidence:** 3
**Soundness:** 3 good
**Presentation:** 3 good
**Contribution:** 3 good

**Summary:**

The paper study unlabeled sample compression schemes for extermal classes and intersection-closed classes.

These classes are of special importance towards solving the sample compression conjecture - every class H has a labeled sample compression of size O((VC(H)). Later on, it was conjectured that it may be true also for unlabeled compressions.

Extermal classes are known to have labeled sample compression linear in the VC - Moran and Warmuth 2016.
Maximum classes and extremal intersection-closed classes have unlabelled compression schemes linear in the VC  - Chalopin, Chepoi, Moran, and Warmuth (2018).

Main contributions:
- Intersection-closed class can be embedded into an extremal intersection-closed class with a linear increase in VC dimension.
- Simplifying the construction for unlabeled compression for maximum classes.
- Showing that every extermal class of VC d, containing a maximum class of VC d-1, has an unlabeled compression scheme of size O(d).


**Questions:**



**Ethics Review Area:**

["I don’t know"]

**Limitations:**

Some sections (in the main body of the paper) can be written in a less technical way, and provide more intuition/explanations of the proofs.

**Strengths And Weaknesses:**

The paper makes nice progress toward showing an unlabeled compression scheme for extermal classes,
which is of great importance in the sample compression literature.

Also, some useful results and techniques are introduced in the paper, that may be helpful in future research in the field, such as Thm 3.4, 3.5, 4.1.

Altogether, I recommend accepting the paper.

---

> ### Author Response · Authors · 2022-08-02
> **Author Response to Official Review**
>
> We thank the reviewer for their positive assessment of the paper, placement of the paper in context of the sample compression literature, and precise summary of the paper’s main contributions.
>
> Following feedback from the four reviewers, we have submitted a revision to the paper, with a colour-coded diff to the original version attached as a supplemental file. Our revisions add to the accessibility of the paper to a general learning theory audience. We’re more than happy to make further improvements to the paper during the author-review discussion.

---

### Author Response · Authors · 2022-08-02
**Rebuttal revision**

We thank all four reviewers for their assessments. We have submitted a **rebuttal revision** of the paper based on the first round of official reviews. The new **supplemental file includes a colour-coded diff** between this version and the original. (Underlined blue text represents new material; strikethrough red text represents removed material; with some approximation due to the $\LaTeX$ package used.)

We have clarified all identified ambiguities, defined background ideas, and included new text throughout to make the paper accessible to a broad audience. We have attempted to implement all requested changes. To make room for these improvements in the main body, we have moved the proof of Theorem 3.4 to a new Appendix A at the end of the paper.

---

### Meta-Review · Area_Chair_nenw · 2022-08-25

**Recommendation:** Accept
**Confidence:** Certain

**Metareview:**

The reviewers agree that this is a solid contribution. Please do revise the paper according to the reviewers comments and the discussion. In particular, the reviewers point out that the presentation can and should be improved: this is important because NeurIPS is a broad conference and the theoretical paper is init should be accessible to the general learning theory community.

**Award:**

No

---

### Decision · Program_Chairs · 2022-09-14

Accept